# Technological Improvements in the Genetic Diagnosis of Rett Syndrome Spectrum Disorders

**DOI:** 10.3390/ijms221910375

**Published:** 2021-09-26

**Authors:** Clara Xiol, Maria Heredia, Ainhoa Pascual-Alonso, Alfonso Oyarzabal, Judith Armstrong

**Affiliations:** 1Fundació per la Recerca Sant Joan de Déu, Santa Rosa 39-57, 08950 Esplugues de Llobregat, Spain; clara.xiol@sjd.es (C.X.); mariaherediaaa@gmail.com (M.H.); ainhoa.pascual@sjd.es (A.P.-A.); alfonsoluis.oyarzabal@sjd.es (A.O.); 2Institut de Recerca Sant Joan de Déu, Santa Rosa 39-57, 08950 Esplugues de Llobregat, Spain; 3CIBER-ER (Biomedical Network Research Center for Rare Diseases), Instituto de Salud Carlos III (ISCIII), 28029 Madrid, Spain; 4Clinical Genetics, Molecular and Genetic Medicine Section, Hospital Sant Joan de Déu, 08950 Barcelona, Spain

**Keywords:** Rett syndrome, Rett-like, NGS, WES, WGS, RNAseq, genetics, MECP2

## Abstract

Rett syndrome (RTT) is a severe neurodevelopmental disorder that constitutes the second most common cause of intellectual disability in females worldwide. In the past few years, the advancements in genetic diagnosis brought by next generation sequencing (NGS), have made it possible to identify more than 90 causative genes for RTT and significantly overlapping phenotypes (RTT spectrum disorders). Therefore, the clinical entity known as RTT is evolving towards a spectrum of overlapping phenotypes with great genetic heterogeneity. Hence, simultaneous multiple gene testing and thorough phenotypic characterization are mandatory to achieve a fast and accurate genetic diagnosis. In this review, we revise the evolution of the diagnostic process of RTT spectrum disorders in the past decades, and we discuss the effectiveness of state-of-the-art genetic testing options, such as clinical exome sequencing and whole exome sequencing. Moreover, we introduce recent technological advancements that will very soon contribute to the increase in diagnostic yield in patients with RTT spectrum disorders. Techniques such as whole genome sequencing, integration of data from several “omics”, and mosaicism assessment will provide the tools for the detection and interpretation of genomic variants that will not only increase the diagnostic yield but also widen knowledge about the pathophysiology of these disorders.

## 1. Rett Syndrome Spectrum Disorders: Clinical Picture

### 1.1. Rett Syndrome

Rett syndrome (RTT, OMIM #312750) is a severe neurodevelopmental disorder characterized by a regression of acquired skills, including purposeful hand use and language, after a normal psychomotor development in the first months of life [1]. RTT has an incidence of approximately 1:10,000–20,000 live female births and is the second most common cause of severe intellectual disability in females [2,3]. RTT was first reported in 1966 by the Austrian doctor Andreas Rett, and in 1983, Bengt Hagberg further described the syndrome in a larger cohort of patients [3].

Although belonging to the same clinical entity, patients with RTT show heterogeneous phenotypes, with varying symptoms and severity. In the classic form, patients display a regression in psychomotor development, partial or complete loss of acquired purposeful hand skills and spoken language, gait abnormalities and stereotypic hand movements, which are the required features to diagnose typical RTT. These symptoms are frequently accompanied by breathing disturbances, bruxism, impaired sleep patterns, abnormal muscle tone, and scoliosis, which constitute supportive criteria [4,5]. It is also common that patients with RTT present with acquired microcephaly and epilepsy [3,4,5].

Presently, the diagnosis of RTT is clinical and follows a set of guidelines published in 2010 [4]. A genetic confirmation typically follows the clinical diagnosis of RTT, but its role is still supportive, since mutations in *MECP2* (the main RTT-associated gene) may cause other phenotypes than RTT, and mutations in other genes have been found in patients with a clinical diagnosis of RTT [4,6]. According to the current diagnostic guidelines, patients who only fulfil two of the four required criteria and five of the 11 supportive criteria are diagnosed with atypical RTT.

There are several atypical forms of RTT. The Zappella (or preserved speech) variant is a generally milder form of RTT characterized by the recovery of some language after the regression, with the ability to speak single words or phrases [4,7]. Other atypical forms of RTT entail more severe phenotypes. The Hanefeld (early-onset seizures) variant is defined by a very prompt onset of epilepsy, and in the Rolando (congenital) variant, there is no clear regression, with the symptoms already being apparent during the neonatal period [4].

Although valid diagnostic guidelines highlight the importance of an apparently normal early development followed by a regression period as a distinct feature for the identification of this disorder, recent reports draw attention towards subtle impairments in motor and communication skills in early stages of development and before the onset of clear RTT features [4,8,9]. Both parents and clinicians have reported feeding difficulties, abnormal crying, and delay in reaching developmental milestones in the first months of life before the clear regression period had begun [10]. Since the identification of these slight symptoms, there has been increasing interest in finding molecular and neurofunctional markers that will enable early detection of RTT.

### 1.2. A Broader Clinical Entity: RTT Spectrum Disorders

Despite the common characteristic features, patients with RTT display a wide range of phenotypic variation that can be appreciated after a thorough clinical characterization. The idea behind the standardized clinical criteria is to provide guidelines to evaluate all traits related to RTT and aid differential diagnosis. Nevertheless, there are patients with many of the distinct features of RTT that do not fulfil the established clinical criteria for either typical or atypical RTT. Currently, the term “Rett-like” is used to describe patients with these overlapping phenotypes, but there are no consensus clinical criteria for a Rett-like diagnosis [11,12,13].

Additionally, recent evidence of genetic heterogeneity behind RTT and RTT-like disorders has become available. Over the past few years, massively parallel sequencing technologies used to diagnose RTT and RTT-like disorders have led to the identification of disease-causing variants in many different genes, some of which are novel and others are traditionally associated with other neurodevelopmental disorders or epileptic encephalopathies with a considerable phenotypic overlap with RTT [11,13,14,15].

In light of the findings regarding the phenotypic and genetic heterogeneity behind this clinical entity, the term “RTT spectrum disorders” can be used to encompass all RTT (typical and atypical) and RTT-like phenotypes [13]. The rationale supporting this idea is that, if the dysfunction of several different genes is causing such similar phenotypes, they must be connected in some way, probably because they share functions or because they are involved in the same molecular pathways [13,16].

## 2. Single-Gene Genetic Testing for RTT Spectrum Disorders

### 2.1. MECP2

The genetic cause of RTT remained unknown until 1999 when it was associated with mutations in the *MECP2* gene, which encodes methyl-CpG-binding protein 2 (MeCP2) [17]. MeCP2 is a multifunctional protein involved in transcriptional regulation, chromatin remodeling, micro-RNA processing, and alternative splicing, modulating gene expression levels both at transcriptional and post-transcriptional levels [18,19,20]. Recent studies show that the loss of MeCP2 alters the expression of many genes, but the effects at an individual gene level are small [1]. This indicates that MeCP2 acts as a global genome regulator of gene expression and chromatin architecture that mediates cellular changes through activation and repression of a great number of genes genome-wide [20,21].

*MECP2* has four exons and produces two different isoforms through alternative splicing. Isoform e1 (498 amino acids) encompasses exons 1, 3, and 4, and has its translation start codon in exon 1 (NM_001110792). Isoform e2 (486 amino acids) encompasses all four exons at the RNA level, but has the translation start codon in exon 2, while exon 1 remains noncoding (NM_004992) [3,22,23]. MeCP2e2 was the form initially described, while MeCP2e1 was not identified until 2004 [22,23]. The two isoforms differ in their N-terminal regions, but share all functional domains [24]. MeCP2e1 is the predominant isoform in the brain, while MeCP2e2 is more abundant in other tissues such as fibroblasts [3,23]. Nonetheless, both isoforms coexist in the brain, where MeCP2e2 has shown a later onset of expression during mouse embryonic development and a more restrictive pattern of expression [24,25]. At a functional level, both proteins differ in binding dynamics, turn-over rates, and interacting partners, suggesting non-overlapping functions of *MECP2* isoforms [26].

According to recent studies, approximately 95% of patients with typical RTT and 75% of patients with atypical RTT (especially those with the Zappella variant) have pathogenic variants in *MECP2* [9,27,28]. So far, 925 different *MECP2* variants, including 535 pathogenic variants and 212 variants of unknown significance (VUSs), have been reported in RettBASE, a specialized database that gathers information related to RTT-related genomic variants [29]. These include missense, nonsense, frameshift, splicing, and intronic variants, and moreover, 280 gross deletions that produce the loss of *MECP2* function have also been described and are included in the Human Gene Mutation Database (HGMD^®^) [30]. However, there are eight recurrent point mutations (missense and nonsense) that account for approximately 50% of all genetically diagnosed cases of RTT (Table 1) [27,29]. Interestingly, mutations specific to MeCP2e1, the predominant isoform in the brain, have been found in RTT patients, whereas there are no mutations specific to isoform e2 known to cause RTT [31].

Since its association with RTT, *MECP2* has been studied by direct sequencing (by Sanger sequencing) and gene dosage analysis (by multiplex ligation-dependent probe amplification (MLPA), qPCR, and FISH) to obtain a molecular confirmation of the clinical diagnosis. At the beginning, the studies did not include exon 1 (because it was thought to be noncoding), but with the description of MeCP2e1, with coding sequences in this exon, it was added to routine mutation screening [23,32]. Taking advantage of the recurrence of the mutations, an electronic DNA microchip was developed to detect seven of the eight most common *MECP2* pathogenic variants in a faster and more economical manner [33].

### 2.2. CDKL5 and FOXG1

Although most patients carry disease-causing variants in *MECP2*, several other genes have progressively been linked to RTT over the past two decades. In 2004, pathogenic variants in cyclin-dependent kinase-like 5 (*CDKL5*) were associated with the atypical form of RTT known as the early-onset seizure variant. CDKL5 is a protein kinase that directly interacts with MeCP2 and mediates its phosphorylation. This phosphorylation modulates the function of MeCP2 in neurons [34,35,36]. MeCP2 also binds *CDKL5* at the DNA level and represses its transcription [37].

In 2008, forkhead box G1 (*FOXG1*) was found to be related to the congenital variant of RTT [38,39,40]. FOXG1 is a brain-specific transcriptional repressor that is coexpressed and colocalized with MeCP2 in the postnatal cortex [40]. FOXG1, like MeCP2, associates with histone deacetylase 1 (HDAC1) to repress transcription [41].

Currently, some authors consider the neurodevelopmental disorders caused by pathogenic variants in *CDKL5* and *FOXG1* distinct clinical entities because of the defining features that differentiate them from typical RTT [42,43]. Nevertheless, the connections and interactions among these three proteins highlight their relationship and indicate that they are in fact involved in common processes, which could explain the overlapping symptoms that arise when the function of any of them is impaired.

When *CDKL5* and *FOXG1* were linked to RTT, single-gene molecular diagnosis techniques were also applied to these genes, particularly in *MECP2*-negative patients with RTT. Combining single-gene approaches in these three genes, about 28% of patients with RTT spectrum disorders were diagnosed with disease-causing variants in *MECP2*, *CDKL5,* and *FOXG1* [44].

## 3. The Revolution of Next Generation Sequencing

### 3.1. Gene panels and Exome Sequencing

The arrival of next generation sequencing (NGS) has allowed us to simultaneously sequence multiple genes in the same experiment [45]. NGS for diagnosis is especially useful in genetically heterogeneous disorders, where many successive single-gene approaches may be more expensive and inefficient. Instead of limiting the scope of the genetic study to one single candidate gene, NGS allows us to extend or redirect a genetic analysis if needed [46].

NGS approaches can target anything from a set of specific genes (gene panels) to the whole genome. Presently, the most widely used approach in a medical diagnostic setting is clinical exome sequencing (CES), which targets all exons of the genes currently known to cause monogenic disorders [44,46,47,48]. Nevertheless, the diagnosis by whole exome sequencing (WES), which targets all exons and canonical splice sites of the ~20,000 known protein-coding genes, is becoming more popular [49,50]. The American College of Medical Genetics and Genomics (ACMG) currently recommends WES as the gold standard of clinical practice in children with intellectual disability (ID), developmental delay, or multiple congenital anomalies, due to the reduction in costs and the increase in diagnostic rate [51].

The overall diagnostic yield when applying WES to patients affected with pediatric rare diseases is 28% on average [46,49,52,53]. Nevertheless, the positive diagnostic rate varies greatly depending on the group of genetic disorders considered [52]. The Deciphering Developmental Disorders (DDD) Study successfully diagnosed 28% of the patients enrolled by applying a combination of WES and exome-focus array comparative genomic hybridization (exome-aCGH) [54]. In a recent systematic review, neurodevelopmental disorders (NDDs), among which we can classify RTT spectrum disorders, were found to have a 23.7% overall diagnostic yield by NGS (22.6% using gene panels and 27.3% using WES). Among NDD subtypes, patients with intellectual disability showed the highest diagnostic yields (28.2%), while patients with autism spectrum disorder (ASD) showed the lowest diagnostic yields (17.1%) [55].

Since a good clinical characterization is critical for variant interpretation, it is not surprising that the best defined clinical entities have the highest diagnostic yields. Therefore, patients with general or imprecise clinical diagnoses, such as ASD or ID, tend to show less successful outcomes than patients with a well-defined, accurate clinical description, such as RTT or Noonan syndrome, despite genetic heterogeneity. Moreover, complex phenotypes that can be caused and modified by several factors, such as ASD, are less likely to be definitively diagnosed than monogenic Mendelian disorders with a clear genetic root.

In terms of technical capacities, NGS technologies are especially sensitive to detecting single nucleotide variants (SNVs) and small insertions or deletions (indels), but copy number variant (CNV) detection is also possible through read depth analysis [46,56]. In fact, the power of CNV detection in WES could be superior to that of low-resolution genomic microarrays [57]. Therefore, WES studies that include CNV analysis usually have a higher diagnostic yield.

Including the parents of the affected child in the WES analysis (trio-WES) has certain advantages. This approach allows us to directly assess the inheritance pattern of candidate variants (de novo or inherited), as well as the phase (whether two variants in the same gene are in the same or in different chromosomes). It also enables directly filtering out rare benign familial variants, which may lead to an up to 10-fold reduction in the number of candidate variants to analyze in families where parents are unaffected [46]. Since de novo mutations are the most common cause of neurodevelopmental disorders such as RTT, the trio-based approach can streamline the genetic diagnosis [58]. Even though the sequencing costs of the experiments are higher when compared to proband-only analyses, the reduction in costs of segregation studies can compensate for this fact. In pediatric rare diseases, diagnostic rates can be increased up to about 40% when applying a trio-WES approach [46].

### 3.2. The Results of NGS Studies of Patients with RTT Spectrum Disorders

Many diagnoses of patients with RTT spectrum disorders come from studies where they were enrolled together with other probands in wider groups characterized as NDD or ID. Nevertheless, there are several diagnostic NGS studies focused only on RTT spectrum patients (Table 2). WES studies of RTT spectrum cohorts have had diagnostic yields around 65% when used as a first-tier diagnostic test and slightly lower when performed after a negative CES result [15,44,59,60,61,62,63]. Gene panels have a higher variation in the diagnostic rates of RTT spectrum patients, strongly depending on the subset of targeted genes and the characteristics of the studied cohort [14,44,64,65,66].

The higher diagnostic yield in WES studies focused only on patients with RTT spectrum disorders, usually above 60% (Table 2), when compared to general NGS data, usually around 28–40%, could be explained due to the thorough phenotypic characterization of the patients included in these studies. As mentioned above, an exhaustive clinical characterization is essential to reach a definite diagnosis by NGS, where many candidate variants can be potentially identified and must be interpreted in the phenotypic context of each patient. Due to the problem of phenotypic heterogeneity, researchers working with RTT spectrum cohorts tend to establish rather accurate criteria for inclusion in the studies and are familiar with comprehensive phenotyping, leading to such positive results (see Appendix A for gene list of RTT spectrum genes). 

NGS has identified more than 90 novel causative genes of RTT spectrum disorders over the past 7 years (Appendix A) [11]. Some of these genes had previously been linked to other well-characterized disorders with overlapping features with RTT, such as Pitt–Hopkins syndrome (*TCF4*), Angelman syndrome (*UBE3A*), or Cornelia de Lange syndrome (*SMC1A*), while others are linked to epileptic encephalopathies (such as *STXBP1* and *GRIN2B*), intellectual disability with epilepsy (*IQSEC2* and *MEF2C*), or other NDDs. Moreover, thanks to NGS, novel possible causative genes for RTT spectrum phenotypes, such as *JMJD1C* and *GABBR2,* have also been identified, [13,14,15,44,60,61,62,63,67,68,69,70,71].

The current list of RTT spectrum genes is especially enriched in chromatin modulators (such as *HDAC1*, *MEF2C*, *NCOR2,* or *SATB2*, and including *MECP2*) and genes involved in synaptic function (such as *GABRB2, GRIN2B, SHANK3, IQSEC2, STXBP1, SLC6A1,* or *SYNGAP1*) [11,62]. These pathways and functions have been found impaired in patients with RTT spectrum disorders, as well as in RTT animal models, and this might be the link between RTT spectrum genes and the reason why patients with RTT spectrum disorders present with overlapping features [72,73].

It is noteworthy that patients with pathogenic variants in several of the abovementioned genes (*TCF4*, *STXBP1*, *SCN2A*, *WDR45, KCNQ2,* and *MEF2C*) present with significant phenotypic overlap with patients with typical RTT and meet the four main established criteria for a diagnosis of typical RTT—gait abnormalities, loss or absence of purposeful hand movements, stereotypies and speech loss, or severe deficit [11]. These six genes are the most frequent causative genes in patients with RTT spectrum disorders, apart from *MECP2*, *CDKL5,* and *FOXG1* [14].

**Table 2 ijms-22-10375-t002:** NGS studies performed on RTT cohorts and their diagnostic yields.

Publication	Type of Genetic Testing	Number of Genes in Test	Number of Patients	Diagnostic Yield
Olson et al., 2015 [59]	Singleton-WES	Whole exome	11	64%
Lucariello et al., 2016 [60]	Trio-WES	Whole exome	21	67%
Lopes et al., 2016 [61]	aCGH and trio-WES	Whole genome (aCGH), whole exome (WES)	19	68.5% (58% due to WES)
Vidal et al., 2017 [44]	Gene panels and WES	17 (custom panel), 4813 (commercial panel), and whole exome (WES)	242 (custom panel), 51 (commercial panel) and 22 (WES)	23% (custom panel), 24% (commercial panel) and 32% (WES)
Sajan et al., 2017 [62]	SNP array-based CNV analysis and trio-WES	Whole genome (CNV analysis) and whole exome (WES)	22	68.4%
Allou et al., 2017 [66]	Gene panel and trio-WES	5 (gene panel) and whole exome (WES)	30 (gene panel) and 2 (trio-WES)	10% (gene panel) and 50% (trio-WES).
Yoo et al., 2017 [63]	Trio-WES	Whole exome	34	67.6%
Iwama et al., 2019 [15]	Singleton-WES	Whole exome	77	61%
Henriksen et al., 2020 [74]	Direct *MECP2* analysis and WES	Whole exome	91	NA

Although the genetic findings of NGS are still validated by molecular genetic techniques in standard clinical practice, this approach is more cost-effective than successive single-gene diagnostic testing. Interestingly, some studies have detected, by NGS, pathogenic variants in *MECP2* that were previously missed by classical molecular genetic testing, and some studies show that NGS can outperform Sanger sequencing in detecting heterozygous changes and mosaic variants [46,59,62,75].

### 3.3. NGS Data Re-Analysis

Nowadays, big efforts are made to improve our variant interpretation capacities. Evidence of the biological consequences of variants of unknown significance (VUSs) is obtained from functional studies, descriptions of disease phenotypes associated with novel genes and variants are stored in comprehensive databases, and the characterization of new functional genomic elements enables the correct interpretation of variants.

Therefore, the frequent update of all these data makes it feasible to reanalyze negative WES cases prior to proceeding with whole genome sequencing (WGS). A recent study showed that 30% of the positive cases solved by WGS could be identified by reanalyzing the WES raw data [76]. Moreover, several studies reported a diagnostic rate after reanalysis of 10.5–15.3% within a period of approximately 1 year after the first analysis [77,78,79,80,81].

The increase in diagnostic success after reanalysis can be due to several reasons. In some cases, new diagnoses are reached because of recent publications of disease-gene associations or particular phenotypes that were not considered when the former analysis was conducted [77,78,80,81]. Another common reason is a revision of the patient’s phenotype by the clinician that can eventually redirect the analysis towards a new set of candidate genes or reconsider variants that were previously detected and dismissed [77,79,80]. Re-classification of a formerly detected variant can also change the result of a WES analysis. For instance, functional and in silico studies can help, over time, to reclassify a former VUS as either a benign or a pathogenic variant or to identify a synonymous variant as a splicing aberration [77]. Finally, a common reason is an improvement in the bioinformatics pipeline or a database update that allows for correct detection or annotation of variants missed in the prior analysis [77,78,79,80].

Considering these outcomes, NGS data reanalysis becomes an interesting diagnostic tool to contemplate in medical genetics until WGS-trio costs decrease and WGS-trio is implemented as ordinary clinical care.

## 4. Future Perspectives of Genetic Diagnosis for RTT Spectrum Disorders

### 4.1. The Bigger Picture—WGS

Exome sequencing has been extremely useful in diagnosing rare diseases, such as RTT. It enables the detection of protein-coding and splice-site variants, which constitute 85% of the known disease-causing mutations [82,83]. Nevertheless, this approach targets only 2% of the genome and has certain limitations, and therefore unsolved exome sequencing cases could potentially be elucidated by WGS.

In addition to the protein-coding and splice-site variants detected by exome sequencing, WGS can also identify several types of non-coding variants that can compromise gene function. Introns, which are not sequenced in exome assays, harbor deep intronic mutations that can increase the activity of cryptic splice sites that otherwise seldom produce splicing events, causing an intron inclusion within the transcript. These aberrant splicing events produce dysfunctional transcripts and may lead to disease. Deep intronic variants have been implicated in the pathogenesis of rare diseases, such as ocular albinism due to *GPR143* malfunctioning and hyperammonemia as a result of *OTC* deficiency [84].

Variants located in regulatory elements (promoters, enhancers, and insulators) may modify gene expression levels by changing transcription factor binding affinities to the DNA sequence. In the 3′ untranslated region (3′-UTR), which contributes to the regulation of gene expression by binding to microRNAs, non-coding variants could affect this binding ability, thus modifying transcript stability [85].

Non-coding variation is progressively gaining acknowledgment, but there are still few reports of the implication of these kind of variants in Mendelian disease phenotypes [86]. This may be partly because these regions are not covered by a regular exome sequencing assay, which is the most common type of genetic test nowadays, so pathogenic variants within these sequences simply have not yet been detected. Moreover, the difficulty in deciphering the effects that these variants may produce and the necessity of functional studies to validate these hypotheses hinder the interpretation and report of these variants.

Another advantage of WGS is its power to detect structural variants (SVs) and complex rearrangements. Since exome sequencing must rely almost exclusively on read depth, it can only confidently detect unbalanced alterations (CNVs). WGS, on the other hand, because of its extensive coverage, enables detection of SVs not only by read depth analysis but also by discordantly aligned paired-end reads and split reads, which allow the detection of SV breakpoints (even at nucleotide resolution) [87]. Thus, WGS can detect balanced SVs (inversions and translocations), besides CNVs and insertions. Furthermore, in the case of duplications, it can distinguish whether they are in tandem or inserted elsewhere in the genome. An interesting study by Gilissen et al. identified disease-causing CNVs and SVs by WGS in nine patients with ID that had been previously missed by a comparative genomic hybridization array (aCGH) [69]. One of these variants was a partial duplication of *TENM3* inversely inserted into *IQSEC2*, which is related to a phenotype of intellectual disability and epilepsy that lies within the RTT spectrum [69].

Furthermore, WGS enables reliable identification of runs of homozygosity (ROH), which are long genomic stretches that display identical haplotypes in both homologous chromosomes. WGS is the best strategy to identify these events, since the non-uniform distribution of WES and SNP-array data complicate the detection of these signals, especially the shorter ones [88]. ROH may denote identity by descent (IBD), which means that the homozygosity originates because the two alleles come from the same common ancestor, indicating some degree of consanguinity [89]. These regions may contain pathogenic recessive variants and are especially relevant when suspecting an autosomal recessive inheritance pattern. ROH might also be indicative of uniparental disomy (UPD) [90]. UPD happens when both homologs of a chromosome pair are inherited from the same parent, and it has implications for disease by causing either the lack or the overexpression of genes affected by gender-specific imprinting and by converting recessive pathogenic variants in a heterozygous state in unaffected carrier parents into homozygous variants causing disease in the proband [90,91]. The detection of long regions of homozygosity in NGS data allows us to uncover UPD events, which have been linked to several cases of NDDs [90].

Finally, WGS uniformity in terms of coverage depth and genotype quality enables the detection of variants previously missed by WES, particularly in GC-rich exons [92,93]. In fact, the proportion of false-positive SNVs is 61% lower in WGS compared to WES [92]. While WES has around a 28% diagnostic rate in patients with ID, WGS has been shown to solve up to 42% of cases, even when only the coding region is studied [69].

Despite the advantages of WGS, it is not always the final answer to reaching a diagnosis [46]. The current issue is that the source of the disease will most likely be blended among the huge amount of data generated, and we may not be able to pinpoint it with our still-limited knowledge.

### 4.2. Multi-Omics

While WES yields 20,000–23,000 variants per individual, WGS reveals 3–5 million [94]. A large proportion of these variants lies outside the coding regions and canonical splice sites, which makes the interpretation of their possible effects challenging. To help prioritize the variants detected at the DNA level, WGS analysis can be coupled with several other technologies focused on studying other molecules, such as RNA sequencing (RNA-seq) and proteomics [94,95].

RNA-seq is an NGS approach that allows us to simultaneously sequence and quantify all transcripts (usually coding transcripts) present in a sample. RNA-seq analyses allow us to not only detect variants in the RNA sequence but also to identify aberrant events that may be caused by variants detected in WGS [96]. Aberrantly expressed transcripts can lead to the identification of both coding variants that trigger nonsense-mediated decay (NMD) and noncoding variants in regulatory regions, such as promoters and enhancers, that hamper transcription. Abnormal splicing events may reveal pathogenic variants in canonical splice sites or within exons or introns that produce dysfunctional transcripts. Finally, finding that one allele is absent among the analyzed transcripts of a given gene (monoallelic expression) helps reconsider the effect that a heterozygous variant in a recessive gene may have if it is the only expressed allele [96,97,98].

Mass spectrometry-based proteomics enable us to identify and quantify all proteins present in a sample (the proteome) at the same time [99]. As with RNA-seq aberrantly expressed transcripts, expression outliers in proteomics may help reprioritize variants detected by WGS within those genes and might indicate that those variants affect protein stability or post-translational modifications [95,96].

Integrating data from genomics and other “omics” is currently known as “multi-omics.” Using multi-omics, several studies have unraveled previously unsolved cases by WGS, WES, or gene panels, increasing diagnostic rates in 10–36% of patients [97,100,101].

The most delicate issue when using multi-omics is the decision of which tissue to study. Since not all transcripts and proteins are produced by all cell types, it would be ideal to select the most relevant tissue according to the condition affecting each patient. In the case of neuromuscular disorders, for instance, muscle biopsies have proved to generate more robust data for RNA-seq analysis compared to blood and cultured fibroblasts [100]. In the case of RTT spectrum disorders, the most disease-relevant tissue would be the central nervous system, which is unfortunately not easily accessible. However, cultured fibroblasts reliably express almost 70% of the disease-related genes registered in OMIM and 70–75% of the RTT-spectrum-related genes according to GTEx data (Figure 1) [96,102]. This demonstrates that a minimally invasive procedure, such as a skin biopsy could provide a sample that, at least to some extent, could be useful for multi-omics analyses to increase the diagnostic yield in RTT spectrum disorders.

### 4.3. Mosaicism

The most widely used NGS technologies for the diagnosis of rare diseases are usually designed to detect genetic variants that affect all the cells of an individual (germline variants). Mosaic mutations (those variants only present in a subset of cells of an individual) are more difficult to recognize and call. Since they are usually found at low frequencies, with a nonspecific analysis they can be confused with technical errors or artifacts [103,104].

The effects of a mosaic mutation on an individual strongly depend on the developmental stage at which the mutation occurs, and consequently, of the number of cells that carry the mutation and the tissues to which they belong [105]. These mutations can happen in the first stages of development and be present in several tissues at different frequencies or relatively late in the process and affect only certain tissues or groups of cells [103,105].

In the former case, a deep coverage NGS approach (at least 200×) may be able to recognize and call the mosaic variant, while in the latter, the mutation will only be detectable if the affected tissue is available for analysis [103]. CES or WES approaches may be too comprehensive to achieve such deep coverage. Therefore, the design of a smaller custom gene panel that contains relevant disease-causing genes with sufficient evidence of pathogenicity for mosaic variants might be a possible strategy to streamline resources. On the other hand, single-cell NGS approaches provide the higher resolution and enable a more shallow sequencing coverage to detect mosaicism, but require access to the tissues relevant for the disease, which is a setback when facing neurological phenotypes [103].

The contribution of mosaic variants to neurodevelopmental disorders is becoming increasingly manifest. A 2014 study using WGS to identify de novo mutations in patients with ID validated seven mosaic pathogenic variants in candidate ID genes (6.5% of the presumed germline variants) [69,104]. Moreover, the study of this cohort found that four presumed de novo mutations in patients with ID were in fact mosaic mutations present at very low frequencies (average of 3.54%) in the blood of unaffected parents [104]. In addition, another recent study found that 6.6% of parents of children with epileptic encephalopathies presented with low levels of mosaicism for the disease-causing variant affecting their children [106]. Therefore, when evaluating the implications of mosaic variants for the genetic diagnosis of neurodevelopmental disorders, we must consider both mosaicism in patients and mosaicism in parents.

#### 4.3.1. Mosaicism in Probands

A mosaic variant in a patient with a neurodevelopmental disorder, such as RTT, could lead to a less specific or a milder phenotype, hindering clinical diagnosis. Furthermore, we must consider that, if the pathogenic variant is present only in the affected tissues (such as the central nervous system), it may be inaccessible and thus remain undetected when sequencing a common peripheral blood sample [105]. Several reports have described male patients with mosaic *MECP2* mutations and different presentations of RTT spectrum disorders, ranging from classical RTT to a mild form resembling the forme fruste [107,108,109,110,111,112,113]. A few studies have also implicated *MECP2* mosaicism in female RTT pathogenesis, typically with a phenotypic presentation according to typical or atypical RTT [113,114].

If detected, a low rate of mosaicism may also complicate interpretations of pathogenicity, questioning if a few mutated cells can have a strong enough effect to cause disease. Nevertheless, mosaic mutations with frequencies as low as 1% have been shown to cause focal cortical dysplasia, an epileptogenic neurodevelopmental malformation [115]. *MECP2* mosaic mutations identified in blood samples have been reported to cause disease from frequencies as low as 6.5% in males and 12.28% in females [113].

#### 4.3.2. Mosaicism in Parents

On the other hand, apparently unaffected parents of children with a neurodevelopmental condition can also carry mosaic variants. The mosaicism may be detectable in many tissues but not cause symptoms if there is a low frequency of the pathogenic mutation insufficient to cause disease or causing very mild subclinical manifestations. On the other hand, there are cases where the mutation arises in germ cells and is not present in any other tissue (germline mosaicism). In these cases, the mutation may seem to be de novo if only blood is tested, even though a high proportion of germ cells carry the variant. In these circumstances, correctly detecting parental mosaicism has important implications for adequate genetic counseling, since it increases disease recurrence risk in subsequent pregnancies [105,116,117].

Several familial cases of RTT have been reported where one of the parents had a germline mosaicism that caused them to have more than one RTT case among their offspring, while they remained asymptomatic [17,117,118,119,120,121]. Although most of these case reports implicate maternal germline mosaicism, a recent study in a large cohort of patients with RTT found paternal germline mosaicism in as much as 23.8% of the fathers studied [113]. As these cases raised the awareness on this issue, the detection of parental germline mosaicism as a cause for RTT spectrum disorders has increased considerably in the past few years. Currently, prenatal diagnosis is indicated in any subsequent pregnancy of families with an individual affected by RTT spectrum disorders, even though the detected disease-causing variant is thought to be de novo [105,116].

### 4.4. Functional Validation of Genomic Variants

The outcome of genomic testing may not always be a conclusive diagnosis. Usually, to reach a final diagnosis directly from the sequencing data, a variant must be identified in a disease-associated gene matching the phenotype of the patient, and this variant should be either a known pathogenic variant previously described and characterized in the literature, or an unknown variant very likely to cause disease [122]. This latter case applies to null variants (nonsense, frameshift, canonical ±1 or 2 splice sites, initiation codon, and single exon or multiexon deletions), which can be assumed to disrupt gene function, when loss of function is a known mechanism of disease for a particular gene [123].

In many other cases, the analysis of sequencing data results in the identification of novel, uncharacterized variants in disease-associated genes, that are classified as VUSs. Additionally, if the analysis is not targeted to previously known disease-genes, it may lead to the identification of genes of unknown significance (GUSs). GUSs are genes with no solid evidence of being disease-causing that may be interesting candidates given their functional roles or molecular interactions.

In order to ascertain the biological consequences of VUSs and reach a definite genetic diagnosis, a functional validation is required [122,124]. A functional assay is an experiment (in vitro or in vivo) that can assess the influence of a VUS on protein function or conformation, and thus help re-classify this variant [125]. In order to re-classify VUSs into either pathogenic or benign variants, the results of functional assays constitute a strong criterion according to the ACMG variant interpretation guidelines [123]. Given the large number of VUSs encountered through genomic testing and the need to validate their functional consequences, the field of “functional genomics” is progressively gaining acknowledgement.

There are diverse approaches to the functional validation of the biological consequences of VUSs [122]. The more extensive, untargeted strategy is the use of multi-omics data in order to find evidence of the malfunctioning at RNA or protein levels of genes carrying VUSs, as discussed above [126]. On the other hand, other functional validation methods are targeted to one or few candidate VUSs previously identified in genomic data. Targeted validation methods include rescue experiments (usually performed with patient-derived cells), where the wild-type allele is introduced to see if the pathogenic phenotype is reverted, and test experiments (usually in model systems), where the VUS is introduced and its consequences assessed [122].

Depending on the type of variant and its predicted biological effect, there are different suitable types of assays [85]. Non-synonymous variants (missense and nonsense) may cause aberrant protein structure and function, leading to decreased gene product levels that can be measured by qRT-PCR or Western blotting. Another possible consequence is erroneous cellular localization, which can be detected by immunocytochemistry assays. To assess the impact of splice-site variants, minigene assays, where the splicing pattern of a subset of exons and introns of a gene is studied, can be used to check the effect on splicing of the candidate variant. Testing the effect of VUSs found in 3′-UTRs and regulatory regions tends to be more complicated. Some assays can be applied though, such as luciferase assays, to compare gene expression levels with and without the candidate variant.

Despite the potential of functional assays to unveil the pathogenicity of VUSs, it must always be taken into account that some biological effects of candidate variants may be tissue or cell-type specific, or they may take place only at a certain developmental stage or under specific environmental conditions [122]. Thus, the choice of a relevant assay and model system (patient-derived material, commercial cell-lines, or animal models, etc.), together with a cautious interpretation of the experimental results, are key to successful outcomes.

Functional validation has been carried out to confirm the pathogenicity of several VUSs associated with RTT spectrum disorders [63,127]. For instance, a specific functional assay demonstrated that a recurrent de novo missense variant in *GABBR2,* which was found in 3 unrelated RTT-like patients, impaired the activity of the mutated protein, thus confirming its pathogenicity [63].

As functional assays can potentially transform a possible diagnosis into a certain diagnosis, it is important to implement functional genomics in a diagnostic setting. The major setbacks for this implementation are high economic costs and long turnaround times [122]. To ensure that the limited resources are spent on the functional validation of the most relevant VUSs, expert clinical geneticists should be involved in the selection of candidate variants. Moreover, as more and more functional studies are performed worldwide, the results of these experiments should be stored in specialized databases, which combined with newly developed machine learning methods, could generate variant pathogenicity predictions with the highest accuracy [128]. The access to this valuable information would streamline variant interpretation and shorten turnaround times in future patients.

## 5. Conclusions

Over the last 20 years, the concept of RTT spectrum disorders has evolved from a monogenic disease, towards a spectrum of overlapping phenotypes caused by pathogenic variants in a great number of genes. The genetic diagnosis techniques applied to this clinical entity have concurrently progressed from single-gene genetic testing to genome-wide approaches and integration of different types of data (Figure 2 summarizes the diagnostic approaches discussed in this review).

The fact that the dysfunction of different interconnected genes can give rise to such similar phenotypes suggests the implication of common pathways in the pathophysiology of the disease, but these mechanisms currently remain unknown. Moreover, the high phenotypic resemblance between patients with pathogenic variants in different genes and the also high phenotypic variability among patients with mutations in the same gene complicate the clinical diagnosis. That is why single-gene genetic testing can be inefficient and the more cost-effective solution is multiple gene approaches enabled by NGS technologies. The ability to reanalyze, extend, or redirect a genetic analysis can benefit the patients and their families by speeding up the diagnostic process. A recent study concluded that if WES or WGS had been performed at symptom onset, genetic diagnoses of NDDs could have been reached more than 6 years earlier [93].

The field of genetics is evolving rapidly and the key to an efficient genetic diagnosis lies in a combination of technological progress and organized knowledge. On the one hand, cutting-edge technologies, such as WGS and multi-omics, are expanding the boundaries of researchers towards more genomic regions and new functional levels. On the other hand, new knowledge is being generated about regulatory regions, gene expression and chromatin organization, which will enable the correct identification and interpretation of pathogenic variants within the huge amount of data generated. In the case of RTT spectrum disorders, with high genetic heterogeneity and significant phenotypic overlap, a thorough clinical characterization remains crucial for assessing the pathogenicity of the identified variants and their relationship with the phenotypes of patients.

## Figures and Tables

**Figure 1 ijms-22-10375-f001:**
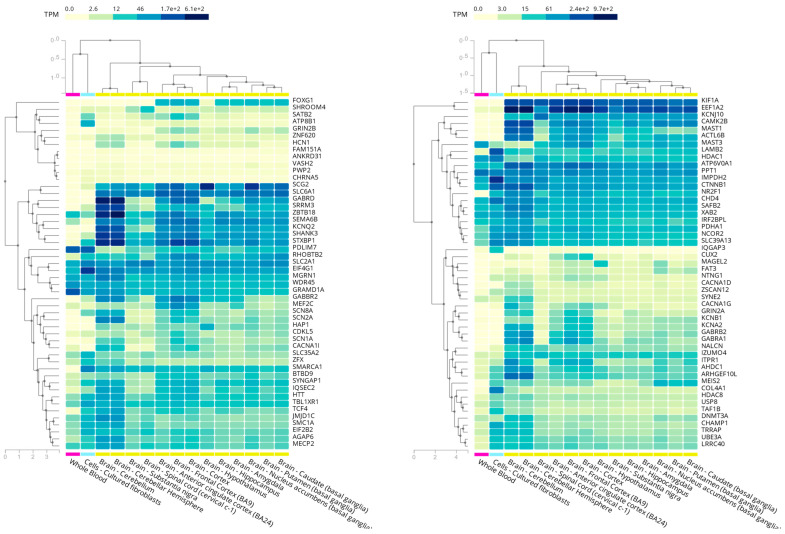
RTT-spectrum-related genes in whole blood, cultured fibroblasts, and different brain regions according to GTEx data (obtained from the GTEx portal on 05/18/21).

**Figure 2 ijms-22-10375-f002:**
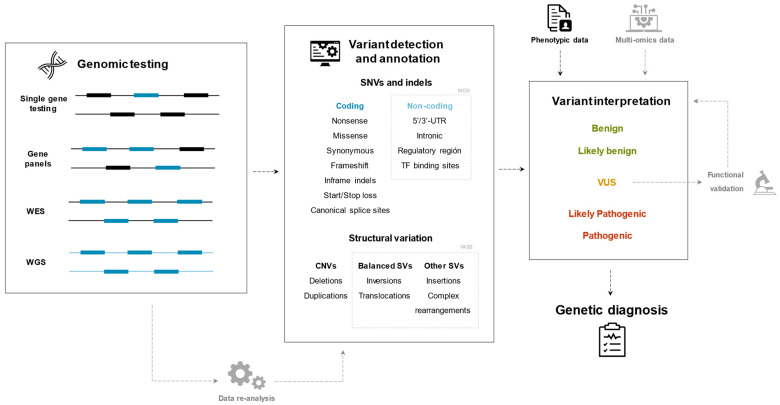
Summary flowchart of the diagnostic approaches presented in this review.

**Table 1 ijms-22-10375-t001:** Most recurrent point mutations in *MECP2* and their frequencies in RTT patients [29].

Coding DNA Variant (NM_004992.4)	Amino Acid Change	Percentage of RTT Patients
c.473C>T	p.Thr158Met	8.74%
c.502C>T	p.Arg168 *	7.57%
c.763C>T	p.Arg255 *	6.64%
c.808C>T	p.Arg270 *	5.74%
c.916C>T	p.Arg306Cys	5.14%
c.880C>T	p.Arg294 *	4.97%
c.397C>T	p.Arg133Cys	4.52%
c.316C>T	p.Arg106Trp	2.79%
		Total = 46.11%

The * in the table represents stop codons according to the current variant nomenclature guidelines of the HGVS (https://varnomen.hgvs.org/recommendations/protein/variant/substitution/).

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
