# Peer review of "Technological Improvements in the Genetic Diagnosis of Rett Syndrome Spectrum Disorders"

_ijms, 2021, doi:10.3390/ijms221910375_

Round 1
Reviewer 1 Report
The manuscript submitted by Xiol and coworkers aims to revisit Rett syndrome and related diseases, especially in the perspective of the novel technological approaches made available for genetics’ diagnostics over the last decade.
It is a very complete work, well written and carefully structured. I do consider that it is of interest for a vast number of the readers of the IJMS journal and contributes positively for the field of Rett syndrome and other neurodevelopment diseases.
Despite my very positive feedback, I considered that most of manuscript (from section 3. and beyond) is written very broadly and not focusing in Rett syndrome (or related diseases) and could eventually targeting intellectual disabilities and neurodevelopment disorders. The authors provide more examples and data about the primary focus of the work. Also, it would be interesting to summarize the approaches for diagnostic in a dedicated picture (e.g., flowchart).
Other minor remarks:
- Please replace the term “Mutation” when referred to disease-causing variant, through the text by this expression or pathogenic variant. The same goes for “mental retardation” that I would use intellectual disabilities.
- Page 3 lines 97 to 99, maybe a reference should be placed here.
- The authors mentioned that eight recurrent point variants in MECP2 account for a significant number of cases, maybe these could be listed or shown in a table.
- Page 4 lines 147 and 148, the authors mentioned that WES will be soon the gold-standard for many rare diseases, it would be important to update that the use of exome or genome sequencing as a first- or second-tier test for children with intellectual disability, developmental delay, or multiple congenital anomalies is strongly recommended for international guidelines (Manickam, K. et al. Exome and genome sequencing for pediatric patients with congenital anomalies or intellectual disability: an evidence-based clinical guideline of the American College of Medical Genetics and Genomics (ACMG). Genet Med (2021). https://doi.org/10.1038/s41436-021-01242-6).
- Page 6 lines 247 to 253, here the authors briefly present some of their one data. However, as these results have not been published yet (no reference) the extractable information is too vague. Thus, I would suggest removing this.
- It would be interesting to incorporate a specific section about the analysis of WES in TRIO. According to the diseases inheritance patterns different filtering strategies are possible, maybe it would be interesting to place this in context.
- About the interesting section addressing mosaicism, I would discuss about the advantages of using a smaller NGS gene panel (custom gene capture) versus more extensive approaches such as CES or WES. Also, could single-cell NGS techniques could bring novelty to this field?
- In terms of the analysis of stretches of homozygosity, these could be indicative (besides UPD) of identity by descent, meaning that there is some degree of parental consanguinity or distant common ancestor. These stretches or runs of homozygosity may be also indicative of a undelaying recessive disease.
Reviewer 2 Report
This review covers all genes that have been reported in Rett syndrome spectrum disorders and discusses the previous and emerging genetic diagnostic techniques used in the field and presents the efficacy of success of each application in improving diagnostic yield.
Major concerns:
The authors mention in the review that they have carried out a WES-trio reanalysis study in patients with features of RTT spectrum disorders who had negative WES results and that these results are pending publication. This data has not been published and therefore the comments the group make are speculation should not be included in this review. The authors should consider the timing of this review and either submit after their reanalysis manuscript has been published or this section should be removed altogether from this review. In addition, it is inappropriate and complete speculation to use the words “we obtained quite promising results,” when referring to one’s own data when it has not been peer reviewed.
The review addresses how new technologies can improve diagnostic rates, but it does not address the process of newly discovered variants/genes and functional validation to confirm their pathogenicity or VOUS using functional genomic studies. This is a major bottleneck in the diagnostic pathway. Further comment on this, how it can be achieved and its consequences should be included. A section dedicated to this is required and further discussion in the discussion section. Conclusions should be discussed on how these new variants are considered pathogenic and VOUS and how (if?) they can (should?) be included in the diagnostic process .
Section 2.1
- “800 different mutations” – Clarification and discussion on exact numbers which are pathogenic and which are VOUS.
- As most cases are caused by mutations in MECP2, more information should be included on the isoforms ie which isoform was initially sequenced (nm number) and which was included later (nm number), how are they different and the spatial and temporal expression of the main isoforms.
Minor concerns:
The introduction does not explicitly mention that diagnosis of Rett syndrome is clinical with defined guidelines (Neul 2010) and there needs some mention of the role of genetic diagnosis in this process.
Language use. The following examples use very strong language which is not supported by references and leads to speculation:
line 65 ‘After thorough clinical characterization, the wide range of phenotypic variation among patients with RTT becomes clearly manifest’ - not clear and could be rewritten for more clarity
Line 72 ‘Additionally, recent evidence of genetic heterogeneity behind RTT and RTT-like disorders has struck researchers’ – The word ‘struck’ is a strong word which requires references as support or reframed.
line 225 : ‘As a result, every single day’: ‘every single day’ is strong language. Is this true and can it be proven?
line 229 : ‘On the other hand, all this work’: ‘all this work’ is strong language. Is this true and can it be proven?
Line 250: “quite promising results” inappropriate use of language especially when referring to unpublished data
Line 420 ‘remarkably’; same as above.
Section 2.2 Line 112: ‘On 2004, mutations in cyclin.....’ Typo change On > In.
In addition this whole sentence requires restructure for more clarity. In addition, if this is a historical account, where the date is supplied for CDKL5 but not for FOXG1.
Line 135 include ‘has’ between ‘(NGS)’ and ‘allowed’
line 147; typo. Change a > the
Line 417 – de novo = italics
